# Association between Altered Blood Parameters and Gut Microbiota after Synbiotic Intake in Healthy, Elderly Korean Women

**DOI:** 10.3390/nu12103112

**Published:** 2020-10-12

**Authors:** Song Hee Lee, Hee Sang You, Hee-Gyoo Kang, Sang Sun Kang, Sung Hee Hyun

**Affiliations:** 1Department of Biomedical Laboratory Science, School of Medicine, Eulji University, 77 Gyeryong-ro, 771 Beon-gil, Jung-gu, Daejeon 34824, Korea; song-1107@naver.com (S.H.L.); yhs1532@nate.com (H.S.Y.); 2Department of Senior Healthcare, BK21 Plus Program, Graduate School, Eulji University, 77 Gyeryong-ro, 771 Beon-gil, Jung-gu, Daejeon 34824, Korea; kanghg@eulji.ac.kr; 3Department of Biomedical Laboratory Science, College of Health Sciences, Eulji University, Seongnam 13135, Korea; 4Department of Biology Education, Chungbuk National University, Chungdae-ro 1, Seowon-gu, Cheongju 28644, Korea; jin95324@cbu.ac.kr

**Keywords:** blood markers, *Blautia*, elderly women, *Ellagibacter*, 16S rRNA, synbiotic drink

## Abstract

Synbiotics intake can alter the composition of intestinal microbes beneficially. We aimed to detect the changes in the intestinal microbiomes of 37 healthy elderly Korean women after the intake of a synbiotic drink. This was a longitudinal study controlled with a temporal series, including a control period of 3 weeks before intake, synbiotic intake for 3 weeks, and a washout period of 3 weeks. Fecal microbiota composition was analyzed by sequencing the V3-V4 hypervariable regions of 16S rRNA. Physical fecal activity increased with improvement in fecal shape. Thirty intestinal bacterial taxa were observed to change only after the intake period. In particular, *Ellagibacter* appeared only after ingestion. In addition, the abundance of Terrisporobacter showed a positive correlation with C-reactive protein, triglyceride. Lachnospiraceae_uc, Eubacterium_g5, and Blautia had a positive correlation with creatinine, whereas PAC001100_g had a negative correlation with creatinine. Short-term (3 weeks) intake of symbiotic organisms changes the composition of the gut microbiota in healthy elderly Korean women.

## 1. Introduction

The life expectancy of Koreans is 83.2 years, but the healthy lifespan (i.e., one without serious illness or injury) was only 64.4 years in 2018 [1,2]. Therefore, the current challenge is to maintain “active longevity” by extending the healthy lifespan and reducing the duration of illness. The human intestinal microbiota play an important role in maintaining gastrointestinal homeostasis and are beneficial for host health [3]. Further, dysbiosis led to an imbalance of beneficial and harmful microorganisms [3]. Gut microbiota dysbiosis causes, and is associated with, several diseases, ranging from localized gastrointestinal disorders (including constipation) and metabolic alterations, to respiratory, liver, cardiovascular, and psychiatric neurological disorders [4]. Using probiotics has been proposed as a potential therapy to restore the balance of healthy human intestinal function and gut bacterial composition [3]. According to the official definition of the World Health Organization (WHO), probiotics are “live microorganisms, which when consumed in adequate amounts, confer a healthy effect on the host” [5].

Although previous studies have showed the positive effects of probiotic consumption on several health outcomes, many of these studies focused on populations with specific pathologies [6,7,8], potentially resulting in side effects [9]. Evidence supporting the health-promoting effects of probiotics in healthy adults is limited and less consistent [10,11,12]. Although many studies have investigated correlations with specific blood markers by targeting a single strain to observe disease associations, the findings of these studies might result from numerous confounding factors [13]. There are also differences in microbial strains, viable cell concentrations, and probiotic product formulations [14], and most studies only controlled some factors affected by these variables. However, a clinical study requires the consideration of all potentially unknown variables. Studies targeting gut microorganisms displaying changes only upon the consumption of target food items are rare [15]. Thus, the aim of this study was to address this limitation and investigate the association between the gut microbiota and blood parameters upon synbiotic consumption.

Recent studies have investigated synbiotics, which are a mixture of one or more probiotics and one or more prebiotics that benefit the host by promoting the survival and colonization of live microbes in the gastrointestinal tract [16,17,18,19]. Synbiotics were developed to overcome the challenges of viability of probiotics; the reason for their use is based on the observed improvements in the number of viable probiotic bacteria during passage through the upper intestine [20].

There are some studies demonstrating the effectiveness of synbiotics [13,21,22,23]. Several studies have shown that the use of synbiotics has superior benefits over probiotics or prebiotics alone. In one study, it was observed that synbiotic intervention results in a significantly different fecal flow bacterial community compared to prebiotic or probiotic alone intervention [21]. In addition, it was much more effective than using probiotics in the satisfaction of bowel habits and overall relief of IBS symptoms [24] and showed an additive effect on constipation symptoms [25]. In animal studies, the use of synbiotics has shown significant anti-inflammatory effects in reducing disease severity, colon damage, and inflammatory mediators, while modulating intestinal metabolites and short chain fatty acids (SCFA) profiles. It has also been found that direct human testing is necessary [26].

The major effects of synbiotics include increasing beneficial gut microbiota and balancing gut microbiota [21,22,23,27,28]. However, no information is available on their impact on the entire phylogenetic structure of the intestinal microbiota in aged people. Therefore, this study observed the beneficial effects of synbiotic intake in healthy elderly women. Further, previous studies on the association between synbiotics and human health have mostly focused on Western populations, with few studies based on the Korean population [29]. In this study, 37 healthy elderly women were given a synbiotic drink for 3 weeks, and various blood and urine parameters were assessed along with fasting blood glucose levels. Furthermore, defecation activity was investigated using a questionnaire and the stool consistency was evaluated under the Bristol stool form scale (BSFS) [30,31]. Fecal microbiota were analyzed through next-generation sequencing of 16S rRNA genes, and the association between the parameters described and the relative abundance and composition of microorganisms was investigated.

## 2. Materials and Methods

### 2.1. Participants and Study Design

This study included 39 healthy women volunteers selected among members of the Miraeseum Seongnam senior complex in Seongnam city, Gyeonggi-do Province, Korea. All participants provided written informed consent and were given the option to withdraw from the study at any time. The protocol was approved by the internal review board (IRB) of Eulji University (IRB No. EUIRB 2019-53). The inclusion criteria were healthy post-menopausal women over 50 (Table 1). The exclusion criteria were: chronic smokers (≥20 cigarettes/day), alcoholism and those who consume 420 g alcohol per week, liver disease, kidney disease, cardiovascular disease, cerebrovascular disease, pancreatitis, cancer, thyroid disease, dementia, Parkinson’s disease, depression, anorexia/bulimia, or multiple sclerosis. Participants were made aware that during the study, those who participate in other clinical research experiments or ingest probiotics, fermented products, and dietary supplements that affect body fat metabolism and intestinal health would be excluded from the study. The participants had to be willing to maintain a usual diet and physical activity levels and were asked to report concomitant medication, adverse effects, or any other comments. We conducted a survey consisting of questions such as family history, past history, and usual diet about the disease (Appendix A
Appendix A).

The study was designed as a longitudinal, panel, pilot study and was controlled with a temporal series (Figure 1). The study included (1) a 3-week control period in which all participants maintained their usual diet and dietary habits, including restrictions on intake of probiotics, prebiotics, and synbiotics, followed by (2) a 3-week experimental period with the consumption of the synbiotic drink (one bottle and two capsules/day), and (3) a 3-week washout period without synbiotic drink intake. Anthropometric measurements and blood, urine, and fecal samples were collected. All participants responded to questionnaires related to basic information such as disease history, use of antibiotics, defecation, and diet. Necessary arrangements were made so the subjects would receive the synbiotic drink every week. The synbiotic was taken after meals to minimize the breakdown of probiotics by gastric acid [32]. Compliance with synbiotic drink consumption guidelines at home was monitored once per week through telephonic interviews. The primary objective of this study was to observe changes in the composition and abundance of gut microbiota, including *Bifidobacterium* and *Lactobacillus,* in elderly Korean women upon consumption of synbiotic drinks. The secondary objective was to observe the association between the blood parameters measured in this study and the bacteria that changed only upon ingestion of the synbiotic drink.

### 2.2. Contents of the Synbiotic Drink

The synbiotic drink products (commercial products, MPRO3) used in this study comprised two formulations: a capsule (2ea) and solution (130 mL). The capsule (2ea) contained *Bifidobacterium animalis* spp. *lactis* HY8002, *Lactobacillus casei* HY2782, and *L. plantarum* HY7712 with crystalline cellulose and lactose. The solution (130 mL) contained 9800 mg of dietary fiber (polydextrose, chicory dietary fiber, and wheat dietary fiber), 3160 mg of fructo-oligosaccharides, 500 mg of xylo-oligosaccharides, and 350 mg of isomalto-oligosaccharide with the same strains as those in the capsule, along with 1000 mg of lactulose powder and milk crude oil. The total bacterial count was 5.0 × 10^9^ colony-forming units (CFUs) for *Bifidobacterium animalis* spp. *lactis* HY8002, 2.5 × 10^9^ CFUs for *L. casei* HY2782, and 2.5 × 10^9^ CFUs for *L. plantarum* HY7712. The ratio of bacteria to capsule and solution was 1:2. The products were obtained from Korea Yakult Co. (Gyeonggi-do, Korea). The nutritional composition of the synbiotic drink is stated in Appendix A. The quality of all products, including harmful microbial levels, were also checked by Korea Yakult Co.

### 2.3. Sample Collection

Information on food consumption, fasting blood samples, and anthropometric measurements were collected at three intervals: before the study (baseline), after the intake period of 3 weeks (after), and at the end of week 6 (washout).

### 2.4. Anthropometric Information

Weight was measured in the fasting state. Height was measured without shoes to the nearest 0.5 cm, and weight was measured similarly, wearing light clothes, to the nearest 0.1 kg. Waist circumference was measured midway between the lower rib margin and the iliac crest using a non-expandable measuring tape to the nearest 0.5 cm. body mass index (BMI) was calculated as the weight (kg) divided by squared height (m^2^) and was interpreted according to the WHO 2000 guideline [3]. A sphygmomanometer was used to measure blood pressure after participants had rested for 5 min in the sitting position.

### 2.5. Blood Sample Collection

After blood pressure measurement, a blood sample was drawn from the antecubital vein in the arm. Blood samples were collected after overnight fasting and abstinence from any medications, tobacco, alcohol, tea, and coffee in EDTA Vacutainer blood collection tubes (Becton Dickinson, Franklin Lakes, NJ, USA) and were mixed. Plasma and red blood cells were separated in the sample via centrifugation (1500× *g*, 4 °C, 15 min). For the measurement of serum biochemical parameters, blood samples collected in EDTA tubes were sent to a hospital (Central Hospital, Seongnam City, Korea) and were immediately analyzed using standard laboratory methods and certified assays.

Biochemical parameters were analyzed in the Central Hospital (Seongnam City, Korea). Glucose (FBG), triglyceride, cholesterol, high-density lipoprotein (HDL), low-density lipoprotein (LDL), LDH, creatinine, alkaline phosphatase (ALP), and C-reactive protein (CRP) were measured using an automated analyzer (Roche Diagnostics, Mannheim, Germany). The remaining blood was stored at −80 °C. All blood test items including our target blood test items and safety parameters are stated in Appendix A.

### 2.6. Urine Sample Collection

Dipstick urinalysis was performed for each urine specimen using a combo stick 10 of a combo stick 720 general urine test strip (DFI, KimJae, Korea). A laboratory technician performed the test within 2 h of sample collection by placing the dipstick in the urine sample according to the manufacturer’s instructions. The results were read within 5 min [33].

### 2.7. Fecal Sample Collection

We provided the participants with a stool container before each visit. The feces were freshly collected (0.25 g) the night before or the morning of the visit. The obtained fecal samples were stored in a domestic refrigerator at 4 °C for less than 4 h before transportation to the laboratory. The sample was transferred to the laboratory, and DNA was extracted. The samples were then stored at −80 °C until analysis.

### 2.8. Microbial Analysis

#### 2.8.1. DNA Extraction

DNA extraction was performed using a QIAamp PowerFecal pro-DNA kit (Qiagen, Hilden, Germany) following the manufacturer’s recommendations. In brief, a 250-mg aliquot of the fecal sample was transferred to a Dry Bead Tube provided in the kit. Next, 800 µL of C1 solution was added, and the sample was vortexed at the maximum speed for 10 min. The rest of the protocol followed the manufacturer’s recommendations. DNA was eluted in a 65-μL C6 elution buffer solution. The extracted DNA samples were stored at −80 °C until library preparation and sequencing [34].

#### 2.8.2. 16S rRNA Amplicon Library Preparation

The fecal DNA samples were analyzed by high-throughput 16S rRNA gene amplicon sequencing analysis using an Ion PGM (Thermo Fisher Scientific Inc., Waltham, MA, USA) next-generation sequencer. The V3-V4 regions of 16S rRNA genes from each sample were amplified using the following adapter sequences, the index sequence, and general-purpose primers: 341F (5′-CCT ACG GGN GGC WGC AG-3′) with a sample-specific 6–8-base tag sequence and 805R (5′-GAC TAC HVG GGT ATC TAA TCC-3′). Polymerase chain reaction (PCR) was performed with the Platinum PCR SuperMix High Fidelity system (Invitrogen, Carlsbad, CA, USA; Thermo Fisher Scientific Inc., Waltham, MA, USA), using 2.5 ng of template DNA and each primer at a final concentration of 50 nM in a 27 μL final reaction volume. PCR was performed under these cycling conditions: 94 °C for 3 min, followed by 30 cycles of 94 °C for 30 s, 50 °C for 30 s, and 72 °C for 30 s. The amplicon libraries were further purified to remove residual primer dimers and any contaminants using the Agencourt AMPure XP DNA purification kit (Beckman Coulter, Brea, CA, USA) following the manufacturer’s instructions. Samples were eluted in 15 μL low TE buffer. The DNA concentration, quality, and amplicon library concentrations were assessed using the dsDNA high sensitivity assay kit on the Qubit 4 fluorimeter instrument (Invitrogen, Carlsbad, CA, USA; Thermo Fisher Scientific Inc., Waltham, MA, USA). The fragment size and quality of the pooled DNA were assessed using an Agilent 2100 Bioanalyzer (Agilent Technologies, Palo Alto, CA, USA). The enriched particles were loaded onto an Ion 554 chip (Thermo Fisher Scientific Inc., Waltham, MA, USA), and sequencing was performed on an Ion PGM according to the manufacturer’s instructions [35,36,37,38].

#### 2.8.3. Analysis of 16S rRNA Amplicon Sequence

The FASTAQ file, the raw data of 16S rRNA sequences, was obtained through Ion Torrent Suite Software version 5.14.1.1. (Thermo Fisher Scientific Inc., Waltham, MA, USA) The 16S rRNA workflow module in EzBio cloud software (ChunLab, Inc., Seoul, Korea) 47 was used to classify individual reads by combining the Basic Local Alignment Search Tool with the curated Greengenes database, which contains a high-quality library of full-length 16S rRNA sequences. The reads were excluded from the analysis if they were shorter than 500 bp or inappropriately paired. Chimeras were identified using USEARCH software (drive5 bioinformatics software and services, https://www.drive5.com USDOE Office of Science (SC), Biological and Environmental Research (BER) (SC-23), Berkeley, CA, USA) and were removed from the sequence data. Sequences were clustered into operational taxonomic units (OTUs) at 97% identity using QIIME’s pick open reference otus.py and the Greengenes 13.5 reference database and UCLUST algorithm.

### 2.9. Statistical Analysis

Statistical analyses were performed using SPSS (version 20.0; SPSS, Inc., Chicago, IL, USA) and GraphPad Prism, version 8.3.1 for Windows (GraphPad software, San Diego, CA, USA). Normality was verified for each outcome and the corresponding non-parametric tests were conducted on outcomes with a skewed distribution. The changes (mean ± SD) in individual biochemical indices and microbiota between the baseline and after the intervention and washout period were assessed with a paired *t*-test or a Wilcoxon signed-rank test according to the data distribution. Differences were considered statistically significant at *p* < 0.05. Spearman’s correlation coefficient (rho, ρ) was used to determine associations between intra-individual similarities of the microbiota after synbiotic drink intake. PCA (Principal component analysis), a linear dimension reduction method, was used to determine major changes at the genus level, and data were compressed into several informational features that allowed for the two-dimensional visualization of sample similarity. Beta diversities were analyzed using the EzBio cloud dashboard (ChunLab, Inc., Seoul, Korea) [39]. PCA was conducted based on the Jensen-Shannon method [40]. The Ion Reporter suite (Thermo Fisher Scientific Inc., Waltham, MA, USA) was used to filter polymorphic variants. Statistical taxonomic comparisons were performed using linear. discriminant analysis effect size (LEfSe) analysis on the Galaxy Hutlab online platform (Hutlab, Boston, Massachusetts, USA; https://huttenhower.sph.harvard.edu/galaxy) [41,42]. LEfSe is a tool developed to find biomarkers between two or more groups using relative abundances, and linear discriminant analysis (LDA) explicitly attempts to model the difference between the classes of data.

## 3. Results

### 3.1. Study Design and Characteristics of Participants

Characteristics of the 37 participants are summarized in Table 1. All subjects were postmenopausal women aged 65.7 years on average, and all were non-smokers. The study protocol is outlined in Figure 1. All subjects made three official visits during the study period for recruitment and screening. The experimental schedule comprised a baseline, ingestion, and washout phase lasting 9 weeks, with 3-week intervals. In addition, we investigated whether a difference in bacteria aligns with the alcohol consumption pattern in Table 1, but we observed no significant difference.

### 3.2. Blood Parameters and Anthropometric Measurements

The results of blood parameters and anthropometric measurements are summarized in Table 2. Glucose levels were significantly decreased by 6% after 3 weeks of synbiotic drink ingestion (*p* = 0.005). Triglyceride (TG), cholesterol, and LDL-cholesterol levels decreased after intake compared to levels at the baseline, but the changes were not statistically significant (*p* = 0.474, 0.733, 0.930, respectively). Creatinine levels were significantly decreased by 15% and ALP level was decreased by 3% after 3 weeks of synbiotic drink ingestion (*p* = 0.025, 0.047, respectively). Body weight, BMI, and blood pressure did not differ significantly among measurements at baseline, after the intervention, and during the washout period (Table 2). Biochemical markers of safety were also measured at baseline, after intake, and after the washout period. (Appendix A). The results showed comparable levels of all safety biomarkers among the women at the study endpoint, and none showed parameters outside the normal range with clinically significant values.

### 3.3. Fecal Assessment

We investigated the mean level of fecal characteristics for 3 weeks before and after the intervention, as well as after the washout period (Appendix A). The average defecation frequency was calculated by the number of defecations per week. Compared with that at baseline, the proportion of subjects with a daily defecation frequency increased from 62% to 70% after the intervention. The average defecation frequency increased from 6.16 to 6.35 times per week. Further, the proportion of women with a BSFS 4 (sausage) form of feces increased from 43% to 68% and then returned to 30% at the washout timepoint (Appendix A) [43,44]. The raw data of the fecal type of the subjects are shown in Appendix A.

### 3.4. Intestinal Microbial Community Analysis

Based on the OTUs, the relative abundances of bacteria at the phylum and genus levels were visualized as a pie chart and bar graph, respectively, as shown in Figure 2. The OTU raw data are provided in Appendix A. After the intervention, the proportions of Firmicutes and Proteobacteria were lower than those detected at pre-intervention (81.3% vs. 85.5% and 1.1% vs. 1.2%, respectively), albeit without statistical significance. However, the proportions of Actinobacteria and Bacteroidetes were significantly higher than those measured at baseline (9.0% vs. 7.8% and 8.5% vs. 5.3%, respectively; *p* = 0.026). Changes in the gut microflora after ingestion reverted to the baseline again during the washout period (Figure 2A).

At the genus level, the relative abundance of *Bifidobacterium* was significantly increased after the intervention (8.4%) compared to that at baseline (5.3%) and then decreased again after washout (6.9%, *p* = 0.017). *Faecalibacterium* and *Fusicatenibacter* increased after the intervention (13.8%, *p* = 0.009; 5.1%, *p* = 0.039) compared to the baseline values (11.1% and 3.7%, respectively), and decreased again after washout (9.2% and 4.1%, respectively; *p =* 0.014). In contrast, *Blautia* and *Eubacterium*_g5 decreased after the intervention (19.3% and 2.9%, respectively) compared to the baseline levels (25.1% and 3.9%) and significantly increased again after washout (23.7%, *p* = 0.005; 3.3%, *p* = 0.008). No significant changes were observed in the relative abundance of other genera, with changes only observed after the intervention (Figure 2B).

We also analyzed the most significant species of *Bifidobacterium* and *Lactobacillus*, both of which are in the synbiotic drink formulation. *B. animalis* and *L. plantarum*, which are in the synbiotic drink, increased significantly after ingestion (*p* = 4.20 × 10^−13^ and *p* = 2.40 × 10^−4^, respectively) and decreased again during the washout period (*p* = 2.00 × 10^−12^ and *p* = 2.30 × 10^−5^, respectively). *Bifidobacterium pseudolongum* increased significantly after ingestion (*p =* 2.50 × 10^−8^) and decreased again during the washout period (*p* = 6.50 × 10^−8^). *L. paracasei* increased significantly after ingestion (*p* = 1.80 × 10^−7^) and then decreased again during the washout period (*p* = 2.00 × 10^−7^). However, the relative abundance of *Bifidobacterium longum* did not change after ingestion and significantly decreased again during the washout period (*p* = 0.035; Figure 2C). *P*-values for all items in Figure 2 are stated in Appendix A.

### 3.5. Temporal Changes in the Intestinal Microbiota

As a complementary and validation method to identify differentially expressed taxa, we applied LDA LEfSe analysis, which generated an LDA score and cladogram for visual representation of the results (Figure 3, Figure 4 and Figure 5).

#### 3.5.1. Comparison before and after the Intervention

We found 64 OTUs based on the LDA scores for the effect size of taxonomic groups with relative abundance ratios before and after synbiotic intake and generated bar graphs (Figure 3). Regarding the intestinal microbial abundance before synbiotic drink intake (baseline), at the family level, Lachnospiraceae and Peptostreptococcaceae showed the highest diversity. At the genus level, we identified 24 genera of Lachnospiraceae. We also identified two prevailing genera of Peptostreptococcaceae, namely *Asaccharospora* and *Peptostreptococcaceae*_uc. Moreover, *Vagococcus* was the most dominant genera that did not belong to the abovementioned families. *Klebsiella*, *Fusobacterium*, and *Peptoniphilus* were also observed (Figure 3B,C).

After ingestion of the synbiotic drink, Bacteroidetes was observed at the phylum level. At the class level, Bacteroidia of Betaproteobacteria and Bacteroidetes showed the highest diversity. At the order level, Pasteurellales, Pseudomonadales, Neisseriales, Flavobacteriales, and Burkholderiales were dominant. At the family level, Pasteurellaceae, Pseudomonadaceae, Neisseriaceae, Flavobacteriaceae, Sutterellaceae, Porphyromonadaceae, Bacteroidales, and Ruminococcaceae were dominant. At the genus level, *Pseudomonas*, *Kluyvera*, *Faecalimonas*, *Neisseria*, *Flavobacteria*, *Caproiciproducens*, *Capnocytophaga*, *Parabacteroides*, *Fusicatenibacter*, and *Faecalibacterium* were predominant. Further, we observed higher abundance at various levels of relative diversity after synbiotic drink intake (Figure 3).

#### 3.5.2. Comparison between the end of Intervention and Washout Period

Sixty-eight OTUs were found and a bar graph was generated based on the LDA scores for the effect size of the taxonomic group with relative abundance at washout and after synbiotic drink intake (Figure 4). After ingestion of the synbiotic drink, there was a prevalence of Actinobacteria at the phylum level, Actinobacteria_c at the class level, Bifidobacteriales at the order level, Bifidobacteriaceae at the family level, and *Bifidobacterium* at the genus level. In addition, Epsilonproteobacteria was prevalent at the class level, Campylobacterales at the order level, Campylobacteraceae at the family level, and *Campylobacter* at the genus level. At the genus level, *Sellimonas and Aggregatibacter* were also prevalent.

At washout after ingesting the synbiotic drink, at the family level, Lachnospiraceae and its genera *Blautia*, *Eubacterium*, *Faecalimonas*, *Eubacterium*, and *Howardella* were predominant. *Terrisporobacter* and *Porphyromonas* genera were also dominant. Finally, at the phylum level, Saccharibacteria_TM7 was dominant, with Saccharimonas_c at the class level, Saccharimonas_o at the order level, and Saccharimonas at the family level (Figure 4B,C). We additionally compared the washout period with the baseline period before synbiotic drink consumption. However, most profiles matched between these two phases and no significantly different bacteria were found in this analysis (Appendix A).

### 3.6. Comprehensive Comparison of Each Time Point and Associations among All Parameters and Microbiota Composition

The intestinal microbial community changes differed in the comparison between baseline and after ingestion and between after ingestion and washout. We found 30 bacteria that changed only after intake, 20 bacteria that showed increased abundance, and 10 that showed decreased abundance (Figure 5B).

At the family level, the increased bacteria predominantly included seven Lachnospiraceae, six Ruminococcaceae isolates, and Sutterellaceae. At the genus level, *Faecalibacterium*, *Fusicatenibacter*, *Faecalimonas*, and *Caproiciproducens* were predominant. Lachnospiraceae was the most dominant family of bacteria, the abundance of which decreased after ingestion. At the genus level, the bacteria that decreased after ingestion at high rates were *Blautia*, *Eubacterium*_g24, and *Eubacterium*_g5. We further assessed the relationship between 16 genera and all parameters, including blood values, which were changed by the intervention (Figure 5C). We plotted a heat map using the Spearman correlation coefficient to evaluate how these variables are related. The relative abundance of the genus *Ellagibacter* of the family Eggerthellaceae was negatively correlated with BMI (rs = −0.331, *p* = 0.045). *Terrisporobacter* of the family Peptostreptococcaceae was positively correlated with CRP (rs = 0.353, *p* = 0.032), TG (rs = 0.365, *p* = 0.026), BMI (rs = 0.493, *p* = 0.002), and body weight (rs = 0.425, *p* = 0.009) and was negatively correlated with high-density lipoprotein cholesterol (HDL-C) (rs = −0.380, *p* = 0.020). *Acutalibacter* of the family Ruminococcaceae was positively correlated with fecal shape (rs = 0.566, *p* = 0.000). *Caproiciproducens* of the family Ruminococcaceae and *Eubacterium*_g5 of the family Lachnospiraceae showed negative correlations with fecal frequency (rs = −0.405, *p* = 0.013 and rs = −0.339, *p* = 0.040, respectively). *Lachnospiraceae*_uc, *Eubacterium*_g5, and *Blautia* of the family Lachnospiraceae were positively correlated with creatinine (rs = 0.352, *p* = 0.033; rs = 0.338, *p* = 0.041; and rs = 0.456, *p* = 0.005, respectively), and *PAC001100*_g of the family Ruminococcaceae was negatively correlated with creatinine (rs = −0.415, *p* = 0.011). The changes in fecal shape and frequency were positively correlated with changes in *Caproiciproducens* and *Eubacterium*_g5 abundance, but were negatively correlated with changes in *Acutalibacter*. The changes in BMI, weight, and TG were negatively correlated with changes in *Terrisporobacter.* Further, significantly positive correlations were observed between creatine and changes in *Lachnospiraceae*_uc (rs = 0.352, *p* = 0.033), *Eubacterium*_g5 (rs = 0.388, *p* = 0.041), and *Blautia* (rs = 0.456, *p* = 0.005), and a negative correlation was found between this parameter and changes in *PAC001100*_g (rs = −0.415, *p* = 0.011).

## 4. Discussion

Countries with high life expectancy are now fighting multiple causes of aging by focusing on diet, hoping to extend human life to 100 years or over. Developing preventive nutritional strategies to maintain or improve the quality of life of an aging population is essential and timely [45,46,47,48]. Several groups are studying diet control in the elderly, and recent reports have focused on the positive relationship between human health and the microbial flora of the human gut [49,50,51,52].

Among the gut bacteria that act as functional organisms in humans, probiotics are worth studying because they have a wide range of beneficial effects on host health and longevity. *Bifidobacteria*, which have been studied by many researchers, decrease as aging progresses and have been an indicator of effectiveness when ingesting probiotics [53,54,55]. However, in our study, the ingestion of a synbiotic drink did not increase the abundance of these bacteria in all subjects. Nevertheless, in those who did not experience an increase, physiological effects beneficial to health, including fecal activity, were observed. Therefore, our results agree with the reports that the mechanisms by which probiotics influence host longevity are not yet clear.

We emphasized the minimization of variables, which is the limitation mentioned by many studies on probiotics. Changes from phylum to genus level were observed through intestinal microbial community analysis, and in this study, relatively different microbial community ratios were observed. To observe the tendency of microbial changes, we further analyzed the intestinal microbes at the time of washout. The results showed changes in certain communities only after ingestion, which then returned to the baseline levels at washout.

We found 30 bacteria in this study that only change when consumed as a synbiotic drink. These 30 bacteria, which change at the after timepoint, are noted only after synbiotic drink consumption. Among them, six genera and several blood index values were observed to have a consistent tendency. *Ellagibacter* abundance in all participants was associated with BMI. *Terrisporobacter* was associated with CRP, TG, BMI, and HDL-C. Lachnospiraceae_uc, Eubacterium_g5, Blautia, and PAC001100_g were associated with creatinine.

The genus *Ellagibacter* of the Eggerthellaceae family did not appear prior to consumption of the synbiotic drink, but detected after consumption of the synbiotic drink. It did not appear again in washout. Urolitin is known as an intestinal microbial metabolite produced from ellagic tannins and foods containing ellagic acid such as walnuts, strawberries, and pomegranates [56,57]. The health benefits associated with these metabolites vary considerably from individual to individual, depending on the composition of the gut microbiota. Urolitin has several properties, including anti-inflammatory, neuroprotective, cardioprotective, “prebiotic-like”, anti-diabetes, anti-obesity, antioxidant and chemopreventive activity, and improved muscle function through various in vitro and animal studies [56]. To date, two genera of bacteria of the human intestine belonging to the family Coriobacteriaceae have been identified as urolithin producers. It is an interesting finding in this study that one of the two bacterial genera, *Ellagibacter*, appears only when ingested in synbiotic drinks. *Ellagibacter*, found at the time point after ingestion of the synbiotic drink, had a negative correlation with BMI, and those with high *Ellagibacter* had lower BMI levels at the time point after ingestion of the synbiotic drink. However, even if there is a correlation between *Ellagibacter* and BMI, no improvement in BMI was observed in our study. In addition, so far, little has been investigated whether *Ellagibacter* correlates with BMI improvement in vivo [58]. In another study, urolithin was divided into three phenotypes, and found that it appeared differently depending on the health status of participants and demographic characteristics such as age, sex, and BMI. These urolithin phenotypes may differ in the human gut microbiota, and whether they should be considered in clinical trials and could be biomarkers associated with other health benefits or disease predisposition, revealed that further research is needed [59]. Therefore, the intake of a synbiotic drink for a short period of time causes *Ellagibacter* to appear, and whether the intake of a synbiotic drink improves BMI is not known in this study, so a follow-up study is needed.

The Peptostreptococcaceae family *Terrisporobacter* decreased after consumption of a synbiotic drink and increased again after stopping consumption. This finding is similar to the findings of studies involving prebiotic administration in neonatal pigs. When comparing the prebiotic intervention group and the control group, *Terrisporobacter* was more prevalent in the control group after prebiotic administration [60]. The mechanism by which consumption of synbiotic beverages decreases *Terrisporobacter* abundance should be studied. At the time point after ingestion of a synbiotic drink, *Terrisporobacter* showed positive correlation with CRP, TG, BMI, and body weight, and negative correlation with HDL-C. Participants with low *Terrisporobacter* abundance had low CRP, TG, BMI, and weight levels. Conversely, HDL-C levels were high. However, although there was a correlation between *Terrisporobacter* and BMI and HDL-C, there was no improvement in this study. Due to the lack of research on *Terrisporobacter*, our results suggest that future studies on *Terrisporobacter* are needed to explore the underlying mechanisms involved in these effects on anthropometric measurements.

Lachnospiraceae_uc, Eubacterium_g5, and *Blautia* decreased only after ingestion of a synbiotic drink, and increased again after stopping intake. PAC001100_g of the Ruminococcaceae family increased only after intake of a synbiotic drink, and decreased again after stopping intake. Other studies have shown that clustering of Lachnospiraceae induced a significant increase in hepatic and mesenteric adipose tissue weight, as well as rat fasting blood glucose levels, and decreased plasma insulin levels and HOMA-β values. It indicates that Lachnospiraceae has an effect on the development of obesity and diabetes [61]. In addition, Lachnospiraceae and *Blautia* were abundant in NAFLD cirrhosis patients [62]. There are not many studies on Eubacterium. Lachnospiraceae_uc, Eubacterium_g5, Blautia and creatinine showed positive correlations. Participants with low levels of Lachnospiraceae_uc, Eubacterium_g5, and *Blautia* also had low creatinine content. PAC001100_g negatively correlated with creatinine, and people with high PAC001100_g levels had low creatinine levels. Although this correlation was found, it was only significant in baseline and after timepoint comparisons. A study that analyzed the composition and function of gut microbiota in fecal samples from bowel transplant recipients compared to 84 patients with chronic kidney disease (CKD) and 53 healthy subjects, showed that it was negatively correlated with clinical markers such as several beneficial genus serums of Lachnospiraceae. It has been shown to be creatinine [63]. Other studies have also shown that intestinal dysbiosis occurs in CKD and can actively contribute to the progression of renal failure [64]. The main characteristic of CKD dysbiosis is an increase in Proteobacteria [65], but an increase in Lachnospiraceae has also been observed [66,67]. Specifically, we identified four strains that affect serum creatinine, which were only altered by consumption of probiotics such as synbiotics or synbiotic drinks. Therefore, these strains changed after ingestion of synthetic beverages for 3 weeks, which was closely related to creatinine regulation.

Overall, significant changes in the intestinal microbial community were only observed after ingestion. When comparing the gut microbiota between baseline and washout timepoint, little difference was found. This is explained by people having a unique microbial community that is always maintained if there is no change in diet and dietary habits. Thus, supplements containing synbiotics can change the gut microbiota even with short-term intake. Some of the blood indicators (FBG, creatinine, ALP) were observed to be improved even during the washout period after the consumption of the synbiotic drink. The results of one study showed an increase in glucose metabolism for several months beyond the period of probiotic consumption. This allows microbes to process dietary polysaccharides that are not digested by human enzymes, which can affect the metabolism of additional glucose into the pool of glucose that can be absorbed by the stomach. In addition, probiotics elicit powerful anti-inflammatory capabilities by inhibiting the NF-κB pathway, which mediates microbial activation of the immune system through toll-like receptors [68]. In studies on creatinine, a decrease in creatinine may be affected by loss of appetite [69], and an increase in creatinine may be due to intake of angiotensin-converting enzyme inhibitors (ACE inhibitors) [70,71]. Other clinical studies have also shown results that extend the effect to several months beyond the period of probiotic consumption. In that study, the effects of probiotic interventions lasted up to 7 years in reducing the risk of atopic eczema [72]. There are currently few reports on the effects of probiotics on blood ALP. ALP is generally higher in patients with liver disease but decreased in this study. Although not a clinical study, one study noted that mineral deficiency in fish was correlated with low alkaline phosphatase activity in plasma or serum [73]. Other studies have shown that lactic acid bacteria and *Bifidobacteria* vary from country to country and dietary effects are also important. In a study examining the nutritional evaluation of Koreans, most Koreans eat kimchi, vegetables, pastes, and pickles every day [74]. A characteristic feature of the Korean diet is that it consumes a lot of fermented foods, including kimchi, a traditional Korean fermented vegetable dish [75]. It has potential probiotic effects such as cholesterol reduction, antioxidant properties, and antimicrobial activity [76]. Most of the participants in our questionnaire were on a vegetable-based diet, including kimchi, and participants were instructed to maintain their usual diet for the entire study period. Additionally, all participants were elderly and sensitive to general dietary management. Therefore, the maintenance of some of the improved blood indices even during the vacation period is explained by the possibility that the Korean-specific diet or efforts to improve the diet against aging of the elderly were involved.

Overall, we found changes in clinical parameters associated with many changes in total microbial function because of synbiotic drink intake for only 3 weeks. Nevertheless, aspects of the study design should be considered when interpreting these results. This study was a longitudinal panel study with a design suitable for observing changes over time in the same population. Moreover, we tried to avoid many unexpected variables. To maintain food freshness and high adherence to the intervention, the food was distributed once per week and continuous monitoring by research personnel was conducted. Along with targeted blood tests, the safety of subjects was assessed based on blood safety parameters. Although the questionnaire was not administered at every visit, since the content of the questionnaire administered at the two visits was different, it was an index to evaluate the participants’ interest in the research and participation compliance.

The primary strength of this study was that it was not a cross-sectional study but a panel study with a longitudinal design, monitoring the same individuals with the same measurement tools through repeated measurements at different time points, which facilitated data analysis before and after the study. Most probiotic studies compare only before and after consumption. Moreover, most studies suggest that it is difficult to determine whether the altered factors are because of ingested probiotics or other variables, such as changes in dietary habits, antibiotic use, and factors in the external environment. In this study, numerous bacteria changed after ingestion compared to levels before the ingestion period. However, during the washout period, we focused on finding bacteria whose abundance returned to levels observed before ingestion (i.e., the bacteria that changed only after ingesting the synbiotic drink). This is because strains not returning to these levels, are not associated with the synbiotic drink effects.

This study cohort comprised elderly women with a higher intake and interest in probiotics; hence, their compliance to the specific dietary conditions was high. Owing to differences in the behavioral radius, lectures were conducted at the senior complex for several days each week, thus helping researchers better understand various factors in this population, such as the frequent use of drugs, supplements, or antibiotics that could interfere with the gut microflora. None of the participants died or dropped out during the 9-week study period. Overall, various measurements and items were measured in the anthropometric and blood index areas, and emphasis was placed on observing the intestinal microbes that changed only after ingesting the synbiotic drink. Future studies on metagenomic sequencing are thus warranted to target and uncover the associations observed.

The major limitations of this study are the relatively small number of participants and the lack of separate control groups. However, because this was a panel study, the sample population remained constant throughout the study. This was therefore considered the baseline control before the start of the study, thus forming a robust control group.

## 5. Conclusions

Short-term (3 weeks) intake of a synbiotic drink improved participants’ satisfaction with their stool shape, bowel activity, and bowel health improvement. We observed that even short-term consumption of a synbiotic drink can alter the intestinal microbial composition of elderly Korean women. In addition, by stopping and observing the intake, we found 30 significant microbes that only changed intake of the synbiotic drink. Continuous consumption of synbiotics is an important prerequisite for this effect. Synbiotics can be a preventive food supplement for chronic diseases that affect healthy older people. Further research is needed to target specific microbial and blood indicators to investigate the combined effects of synbiotics, probiotics, and prebiotics, along with the causal and clinical effects that are responsible for these changes.

## Figures and Tables

**Figure 1 nutrients-12-03112-f001:**
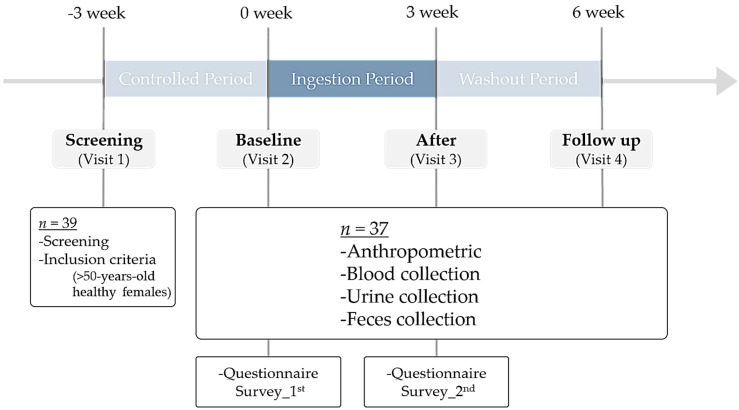
Study flowchart.

**Figure 2 nutrients-12-03112-f002:**
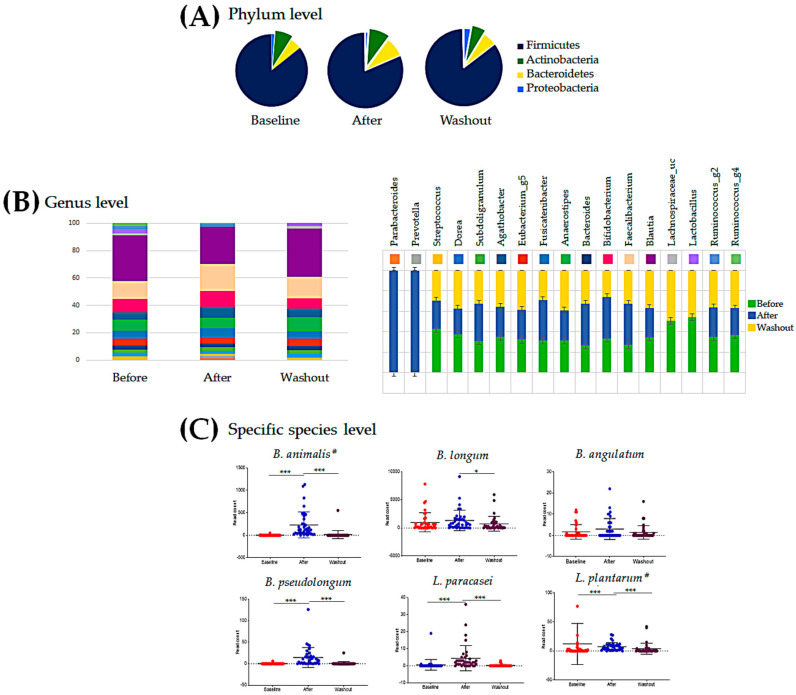
Relative abundance of intestinal microbial phyla (**A**), genera (**B**), and species (**C**) in 37 participants based on 16S rRNA analysis. Numbers indicate the relative contribution (percentage) of each microbial level. * *p* < 0.05, *** *p* < 0.005 (paired *t*-test and Wilcoxon rank-sum test). In (**C**), Crosshatch (^#^) is indicated for the bacteria contained in the synbiotic drink.

**Figure 3 nutrients-12-03112-f003:**
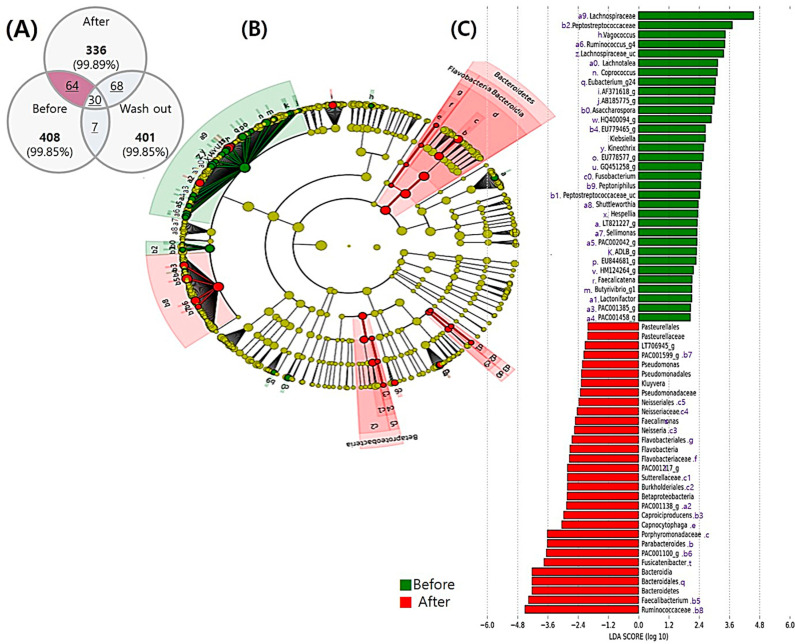
Cross-taxa profile of intestinal microflora in all subjects (*n* = 37) before and after synbiotic ingestion. (**A**) Venn diagram for the distribution of operational taxonomic units (OTUs) at each time point. (**B**) Cladogram showing phylogenetic relationships among taxa with statistically significant differences between time points. (**C**) Linear discriminant analysis effect size (LEfSe) showing the contribution of different bacteria to the differences in the before and after period. Green color indicates taxa enriched before ingestion, and red color indicates taxa enriched after synbiotic ingestion.

**Figure 4 nutrients-12-03112-f004:**
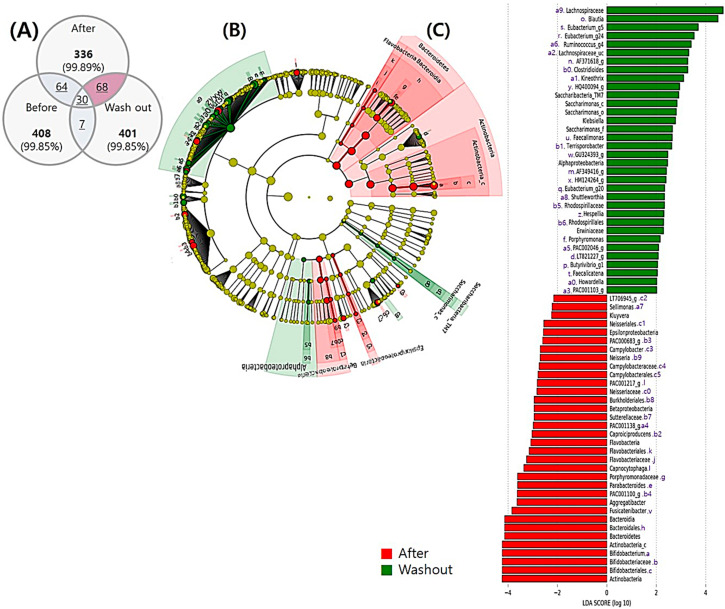
Cross-taxa profile of intestinal microflora in all subjects (*n* = 37) after synbiotic ingestion and during washout. (**A**) Venn diagram for the distribution of operational taxonomic units (OTUs) at each time point. (**B**) Cladogram showing phylogenetic relationships between taxa with significant differences between time points. (**C**) Linear discriminant analysis effect size (LEfSe) showing the contribution of different bacteria to the differences between the after ingestion and washout periods. Green color indicates taxa enriched after ingestion, and red color indicates taxa enriched in the washout period.

**Figure 5 nutrients-12-03112-f005:**
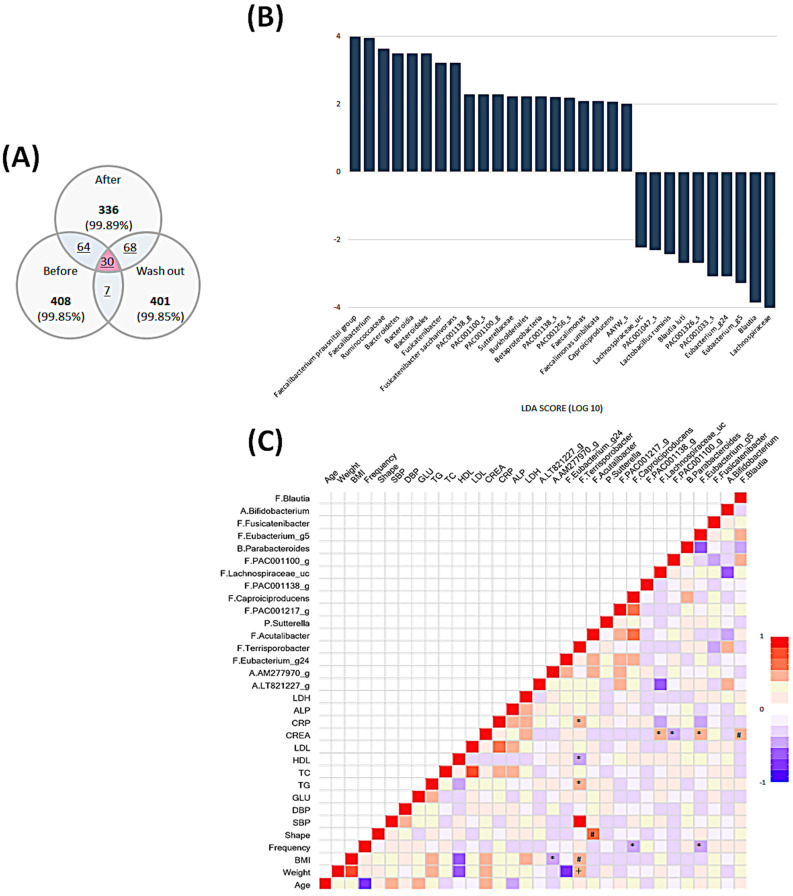
Cross-taxa profile of intestinal microflora and biological parameters in all subjects (*n* = 37) compared among the three periods (before, after, and washout). (**A**) Venn diagram showing the distribution of operational taxonomic units (OTUs) for each time point. (**B**) Based on the linear discriminant analysis (LDA) score, 30 significantly altered bacteria represent an increased and decreased group, including 16 genera with significance and tendencies toward significance. (**C**) Heat map of the Spearman rank correlations between biological and gut microbial outcomes. Red indicates a positive correlation and blue indicates a negative correlation. The heat map shows the first letter of the phylum-level bacteria and genus-level bacteria in order. A, Actinobacteria; F, Firmicutes; P, Proteobacteria; B, Bacteroidetes. * *p* < 0.05, # *p* < 0.01, and † *p* < 0.0005 (after false discovery rate correction). Abbreviations: BMI, body mass index; SBP, systolic blood pressure; DBP, diastolic blood pressure; GLU, glucose; TG, triglyceride; TC, total cholesterol; HDL, high-density lipoprotein; LDL, low-density lipoprotein; CREA, creatinine; CRP, C-reactive protein; ALP, alkaline phosphatase; LDH, lactate dehydrogenase.

**Table 1 nutrients-12-03112-t001:** Clinical characteristics at baseline of the 37 subjects involved in the study.

Variables			*N*	Mean (SD) or Frequency (%)
Age (Years)				
	50–59		11	57.9 (1.2)
	60–69		15	63.3 (3.2)
	70–79		11	73.5 (3.0)
	Total		37	65 (6.7)
Menopausal transition	Menstruating		0	0
	Post-Menopausal		37	100%
Smoking	No (Non-smoking)		37	100%
	Yes (Smoking)		0	0
	Ex-smoker		0	0
Alcohol	No		29	78%
	Yes		8	22%
Alcohol frequency of consumption	Alcohol (Yes)	Daily	5
		Weekly	1	
		Occasionally	2	

**Table 2 nutrients-12-03112-t002:** Changes of anthropometric measurements, biochemical parameters, and fecal characteristics at each timepoint ^1^.

Variables		Baseline	After	Washout	*p*-Value	
					Baseline vs. After	After vs. Washout
Anthropometric measures						
	Weight (kg)	55.27 (7.33)	55.56 (7.29)	55.80 (7.44)	0.221	0.08
	BMI (kg/m^2^)	23.02 (2.92)	23.16 (3.00)	23.22 (3.00)	0.253	0.744

Blood parameters						
	Systolic blood pressure (mmHg)	115.92 (13.84)	116.30 (11.51)	117.73 (15.50)	0.800	0.508
	Diastolic blood pressure (mmHg)	73.54 (7.77)	74.22 (6.61)	74.76 (7.42)	0.606	0.611
	FBG (mg/dL)	94.21 (16.40)	88.95 (15.26)	88.89 (14.21)	0.005	0.974
	Triglyceride (mg/dL)	155.22 (126.26)	150.95 (69.63)	159.11 (83.25)	0.474	0.391
	Cholesterol (mg/dL)	206.73 (42.54)	205.38 (45.20)	200.95 (41.40)	0.733	0.370
	HDL cholesterol (mg/dL)	54.30 (10.42)	54.43 (12.08)	53.68 (11.78)	0.897	0.916
	LDL cholesterol (mg/dL)	121.05 (35.74)	120.76 (43.38)	115.54 (35.65)	0.930	0.294
	Creatinine (mg/dL)	0.92 (0.28)	0.78 (0.42)	0. 56 (0.10)	0.025	0.768
	CRP (mg/dL)	0.15 (0.23)	0.12 (0.13)	0.15 (0.17)	0.317	0.228
	ALP (U/L)	75.32 (17.48)	73.24 (16.25)	73.68 (15.41)	0.047	0.181
	LDH (U/L)	182.43 (27.13)	181.81 (27.50)	167.62 (28.12)	0.239	0.000
Fecal characteristics ^2^	Shape	4.3 (0.9)	4.0 (0.8)	4.4 (1.1)	0.110	0.083
	Frequency	6.16 (1.25)	6.35 (1.22)		0.346	

^1^ All values are presented as mean (±SD). ^2^ Fecal characteristics are based on the standard Bristol stool level. Abbreviations: BMI, body mass index; FBG, fasting blood glucose; CRP, C-reactive protein; HDL, high-density lipoprotein; LDL, low-density lipoprotein; ALP, alkaline phosphatase; LDH, lactate dehydrogenase.

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
