# Peer review of "Association between Altered Blood Parameters and Gut Microbiota after Synbiotic Intake in Healthy, Elderly Korean Women"

_nutrients, 2020, doi:10.3390/nu12103112_

Round 1

Reviewer 1 Report

Major comments:

  • The study design makes it impossible to distinguish the individual effects of the prebiotic and the probiotic.
  • The number of participants (39 women) seem to be a limited number in order to extrapolate the conclusion to the Korean women population.
  • A full study of the dietary habits of the participants would have been necessary.
  • The methods for the determination of blood parameters must be explain in the material and methods section.
  • In conclusions section, authors state that the intake of the symbiotic during three weeks improved the fasting levels of glucose, TG, total cholesterol, LDL-cholesterol, creatinine, ALP, and LDH levels. However, table 2 only showed a significant reduction of fasting blood glucose, and a slight decrease in ALP activity. On the other hand, it is uncertain that a decrease of less than 3% in glycaemia could be considered an improvement in healthy women. In fact, considering these results, it is not clear that the symbiotic could have a significant impact on elderly.

Minor changes

  • Line 35: Keywords: Please, change “blood indices” to “blood markers”.
  • Line 38: “life expectancy of Koreans” or “life expectancy of Korean women”?.
  • Line 88. Material and Methods: Exclusion criteria included all type of chronic disease??.
  • Line 91. Material and Methods. Could authors define “a usual diet”?.
  • Line 93. Table 1: Why male frequency is 65%
  • Line 96. Material and methods: Could authors explain what type of prebiotics were exclude in the diet during the basal control period?. It included the restriction of fiber intake?
  • Line 98. Why authors decided an experimental period of three weeks?. Could they provide some bibliography that support this decision?.
  • Line 235. Result: Table 2. Please, reviewer the values of Systolic Blood Pressure.

Author Response

Please see the attachment* Reviewer 1. docx (word file): Response to reviewer's comment

Sincerely yours,
Song-Hee Lee

Reviewer 2 Report

The work presented by Song Hee Lee and colleagues showed the association between altered blood parameters and gut microbiota after synbiotic intake in elderly Korean women. This study is novel, and the manuscript is well written. However, I just have some minor concerns. • In lines 102-103, please explain the rationale/specific reference of this statement “The symbiotic was taken after breakfast or after dinner to minimize the death of probiotics due to gastric acid”. • Line 225, please indicate the p-value for changes of each blood parameters (like triglycerides, cholesterol, and LDL-cholesterol). • Lines 225-228, please indicate which groups you are comparing here. • In Table 2, it will be better if the authors can give the exact p-value of all individual comparisons. • Line 253, although authors didn't find some comparisons are not statistically significant, it is important to show the p-value for all individual comparisons. • In Figure 2A, please indicate which pie chart represents what. • In lines 292, 321, these sentences are incomplete. • Line 326, the authors can show this data in the supplementary figure/table.

Author Response

첨부 파일을 참조하십시오* 검토 자 2 . docx (단어 파일) : 리뷰어의 댓글에 대한 응답

감사합니다.
이송희

Reviewer 3 Report

This study reports on the clinical effects of a synbiotic drink containing both probiotic and prebiotics in healthy, elderly Korean women and characterizes intestinal microbiome population shifts.

The authors used rigorous methodological techniques to obtain data and took steps to eliminate bias. However, the scientific contribution of this paper is of low impact considering the population studied and the results. Particularly, that this synbiotic drink was used to provide clinical benefit to individuals who were already relatively healthy based on body weight and blood biomarkers, leading to very minor clinical changes. Nonetheless, the microbiota changes are of interest and the overall results are worth reporting.

I recommend English editing services to clean up clunky sentences which are difficult to interpret. For example, Line 95-96, “and were restricted to the intake of probiotics and prebiotics” which sounds like this is all they could have, when you are trying to say the opposite. Line 474-476 is also poorly worded and with poor syntax. These are a couple of examples but there are many sentences throughout the paper which are convoluted.

Symbiotic is accidentally used in place of synbiotic in many instances.

The title should more accurately reflect the population by stating “…synbiotic intake in healthy, elderly Korean women”. It would be assumed that a population with some sort of chronic disease would gain greater clinical benefit. As such, stating “healthy” is important as it is typically not assumed with elderly subjects.

It is unclear whether the benefits of the synbiotic drink are due to the fiber and carbohydrate composition of the drink (prebiotic) or the probiotics themselves. While rational was briefly provided in the introduction, this should be elaborated upon by discussing specific investigations which illustrate how prebiotics in isolation and probiotics in isolation are less effective than their combination.

It was mentioned how food intake was documented. These data were not presented. Nutritional analysis in a supplementary table would be important to account for inter-individual variations in microbiome populations. A high-fiber, low meat diet would reflect a very different microbiome population compared with low-fiber high-meat diet. Perhaps a lack of large clinical changes could even be explained if these subjects were already eating a relatively higher-fiber diet (prebiotic).

In table 1, Male is listed as N of 0 but its frequency is 65%. It should be 0%.

In line 112-115, the percent of each bacterial population in the supplement would be of value.

In line 122-123, you state that heavy metals and harmful microbes were tested for. What were the results of these tests? I am not looking for anything extensive, just that it is free of heavy metals and harmful microbes, which may be better to say outright.

Line 217, subjects are postmenopausal, not menopausal.

Alcohol consumption is discussed. With a significant portion of subjects being daily drinkers. Were there inter-individual differences in bacterial populations in these individuals compared to weekly and occasional users?

In table 2, Diastolic BP for “after” has an incorrect value.

In table 2, Frequency is missing a value for washout.

In the discussion section, it would be good to concisely restate the major bacterial populations that changed in the beginning. While you discuss some of these changes throughout the discussion, the relevance to your results are not always clear.

Line 402-416 is a disjointed paragraph. BMI is discussed and then ellagic acid metabolism is brought in mid-paragraph out of context. What is the relevance of ellagic acid metabolism to Ellagibacter and BMI?

Lines 477-479 it is stated “However, if steady intake is not continued, the baseline conditions will be quickly restored; thus, steady intake is necessary to obtain health benefits”. However, your results contradict this statement, as blood glucose, creatinine, and ALP remained reduced during the washout period. This sentence should not be stated as such or revised to accurately reflect the results.

In your conclusion, you discuss the association with bacterial shifts and changes in BMI and CRP, etc. However, considering BMI did not significantly reduce, and that CRP did not significantly change, these conclusions are not appropriate. If secondary analysis was conducted on subjects that had these bacterial shifts and their BMI/CRP changed, this should be clearly stated and contrasted with other subjects, and perhaps included as a supplementary table.

Author Response

Please see the attachment* Reviewer 3. docx (word file): Response to reviewer's comment

Sincerely yours,
Song-Hee Lee

Round 2

Reviewer 1 Report

Dear authors, I appreciate the review and the corrections in the manuscript that has been improved this new version

Author Response

I didn't attach the Author's Reply to the Review Report file because there were no additional requirements in the Comments and Suggestions for Authors field.

Sincerely yours,
Song-Hee Lee

Reviewer 3 Report

The authors made an effort to address issues stated and clarity is improved. Most issues are resolved, however, not all comments were fully addressed.

In my original comment, “It is unclear whether the benefits of the synbiotic drink are due to the fiber and carbohydrate composition of the drink (prebiotic) or the probiotics themselves. While rational was briefly provided in the introduction, this should be elaborated upon by discussing specific investigations which illustrate how prebiotics in isolation and probiotics in isolation are less effective than their combination.” The authors attempt to address these issues in line 61-69, however, it does not address the main thesis of my question, that being differentiating the difference between prebiotics, probiotics and their combination as a synbiotic. The authors however provide rationale privately in their comments to me. This rationale is what I am looking for in the main text of the document to be explicitly stated, not just privately to me. Explicitly mentioned should be the comment and study “The synbiotic intervention fostered a significantly different fecal stream bacterial community than did either the prebiotic (P = 0.032) or the probiotic (P = 0.001) intervention alone....”.

The same is true for my original comment “Lines 477-479 it is stated “However, if steady intake is not continued, the baseline conditions will be quickly restored; thus, steady intake is necessary to obtain health benefits”. However, your results contradict this statement, as blood glucose, creatinine, and ALP remained reduced during the washout period. This sentence should not be stated as such or revised to accurately reflect the results.”. Again, authors privately provide rationale for this statement, however, do not fully address this in the main text of the manuscript. Authors should explicitly discuss why these values remained changed in washout. State these items within context in the discussion “The results of one study showed an increase in glucose metabolism for several months beyond the period of probiotic consumption. This allows microbes to process dietary polysaccharides that are not digested by human enzymes, which can affect the metabolism of additional glucose into the pool of glucose that can be absorbed by the stomach.”

“Almost Koreans eat kimchi, vegetables, paste, and pickles every day. Participants were instructed to maintain their usual diet. In addition, all the participants were elderly and were sensitive to the usual diet management. Therefore, the maintenance of some of the improved blood indices even during the washout period is explained by the possibility that a Korean-specific diet or efforts to improve the diet against aging of the elderly were involved.”

In my original comment of the conclusion, it is stated “In your conclusion, you discuss the association with bacterial shifts and changes in BMI and CRP, etc. However, considering BMI did not significantly reduce, and that CRP did not significantly change, these conclusions are not appropriate. If secondary analysis was conducted on subjects that had these bacterial shifts and their BMI/CRP changed, this should be clearly stated and contrasted with other subjects, and perhaps included as a supplementary table.” The authors do not fully address the primary criticism, that as a reader, it is entirely unclear that these associations existed based on the data presented. A figure which shows Pearson correlation with BMI and CRP with these bacteria is really needed to suggest such an association.

Author Response

* Reviewer 3. docx (word file): Response to reviewer's comment

Sincerely yours,
Song-Hee Lee
